# Automatic Plaque Removal Using Dual-Energy Computed Tomography Angiography: Diagnostic Accuracy and Utility in Patients with Peripheral Artery Disease

**DOI:** 10.3390/medicina58101435

**Published:** 2022-10-11

**Authors:** Byeong-Ju Koo, Jung-Ho Won, Ho-Cheol Choi, Jae-Boem Na, Ji-Eun Kim, Mi-Jung Park, Sa-Hong Jo, Hyun-Oh Park, Chung-Eun Lee, Mi-Ji Kim, Sung-Eun Park

**Affiliations:** 1Department of Radiology, Gyeongsang National University School of Medicine and Gyeonsang National University Hospital, Jinju 52727, Korea; 2Department of Thoracic and Cardiovascular Surgery, Gyeongsang National University School of Medicine and Gyeonsang National University Hospital, Jinju 52727, Korea; 3Department of Preventive Medicine, Institute of Health Sciences, Gyeongsang National University School of Medicine, Jinju 52727, Korea; 4Department of Radiology, Gyeongsang National University School of Medicine and Gyeonsang National University Changwon Hospital, Changwon 51472, Korea

**Keywords:** peripheral artery disease, automatic plaque removal, dual-energy computed tomography angiography, digital subtraction angiography

## Abstract

*Background and Objectives*: This study aimed to evaluate the utility and accuracy of dual-energy automatic plaque removal (DE-APR) in patients with symptomatic peripheral arterial disease (PAD) using digital subtraction angiography (DSA) as the reference standard. *Materials and Methods*: We retrospectively analyzed 100 patients with PAD who underwent DE computed tomography angiography (DE-CTA) and DSA of the lower extremities. DE-CTA was used to generate APR subtracted images. In the three main arterial segments (aorto-iliac segment, femoro-popliteal segment, and below-the-knee segment), the presence or absence of hemodynamically significant stenosis (>50%) and calcification was assessed using the images. CTA data were analyzed using different imaging approaches (DE-standard reconstruction image (DE-SR), DE-APR maximum intensity projection image (APR), and DE-SR with APR). *Results*: For all segments evaluated, the sensitivity, specificity, and accuracy for detecting significant stenosis were 98.16%, 81.01%, and 89.58%, respectively, with DE-SR; 97.79%, 83.33%, and 90.56%, respectively, with APR; and 98.16%, 92.25%, and 95.20%, respectively, with DE-SR with APR. DE-SR with APR had greater accuracy than DE-SR or APR alone (*p* < 0.001 and *p* < 0.001, respectively). When analyzed based on vascular wall calcification, the accuracy of DE-SR with APR remained greater than 90% regardless of calcification severity, whereas DE-SR showed a considerable reduction in accuracy in moderate to severe calcification. In the case of APR, the degree of vascular wall calcification did not significantly influence the accuracy in the aorto-iliac and femoro-popliteal segments. DE-SR with APR achieved significantly higher diagnostic accuracy for all lower extremity segments in evaluating hemodynamically significant stenosis in patients with symptomatic PAD and transcended the impact of vascular wall calcification compared with DE-SR. *Conclusions*: APR demonstrated favorable diagnostic performance in the aorto-iliac and femoro-popliteal segments, exhibiting good agreement with DSA even in cases of moderate to severe vascular wall calcification.

## 1. Introduction

Peripheral artery disease (PAD) is chronic arterial occlusive disease of the lower extremities caused by atherosclerosis, which is one of the most important processes in the pathogenesis of cardiovascular disease. Therefore, it is also associated with cardiovascular morbidity and mortality, and presents a significant public health problem [1,2]. PAD may be asymptomatic, but it may present with symptoms such as intermittent claudication, atypical leg pain, and critical limb ischemia. Smoking and diabetes are two major risk factors for PAD. In patients with coexisting PAD and diabetes, the lesions are more often multilevel, more distal arteries are affected, and the arterial walls are usually more calcified [3,4]. Imaging is critical for the evaluation and planning of endovascular or open surgical intervention in patients with PAD [5,6]. This is because in the case of PAD, accurate evaluation of the site, length, and severity of vascular involvement is important for optimal follow-up treatment [7]. Digital subtraction angiography (DSA) is considered the gold standard for evaluation of patients with PAD [5,8]; however, DSA is invasive and has potential complications [9]. Duplex Doppler ultrasound is a readily available non-invasive imaging modality but is operator- and patient-dependent, and finding a suitable acoustic window can be difficult [10]. Thus, computed tomography angiography (CTA) is widely applied in the screening and management of lower extremity arteries and vessels due to its safety, non-invasiveness, and ease of operation compared with DSA, as well as its shorter time and fewer contraindications compared with magnetic resonance angiography (MRA) [11,12,13]. CTA’s sensitivity and specificity are reported to be comparable to those of DSA [14]. The data in these reports are the general results of comparisons with DSA in the aorto-iliac segment, femoro-popliteal segment, and below-the-knee segment (AIS, FPS, and BTKS, respectively). However, a major drawback of CTA is that it cannot reliably distinguish between attenuation due to high intravascular iodine concentration and attenuation due to calcium [15]. Therefore, CTA has limited value in the evaluation of widely calcified (particularly BTKS) arteries [10] and conventional bone removal requires time-consuming post-processing [13].

Dual-energy CTA (DE-CTA) enables bone and intraluminal plaque removal from angiographic datasets based on spectral differentiation that separates iodine from calcium, thereby ideally generating a true CTA luminogram [16]. Accurate and time-effective assessment of even sclerosing lesions in PAD by referring to three-dimensional (3D) reconstructed images would render CTA more comparable to MRA and DSA and increase its utility in routine practice [17,18]. Among the available modalities, multiplanar reconstruction maximum intensity projection (MIP) generation has been shown to be less time-consuming than standard CTA scans [19] and provides better visualization and characterization of vascular features. Clinical studies with DE-CTA have revealed promising results [3,14,19,20,21].

This study aimed to explore the utility and accuracy of DE automatic plaque removal (DE-APR), including automatically generated MIPs, in patients with symptomatic PAD, with DSA serving as the reference standard.

## 2. Materials and Methods

### 2.1. Participants

Patients who were referred for runoff DE-CTA of the pelvis and lower extremities due to clinical symptoms of PAD between September 2019 and August 2021 were retrospectively selected for this study. The study included patients with additional DSA of the same body region within 30 days of the DE-CTA scan.

Patients with contraindications to intravenous administration of iodine contrast agent (CA), those with renal dysfunction or failure, those who did not undergo DSA of lower extremity arteries within 30 days after the CT scan, those who underwent surgical or conservative treatment, or those with a reasonable diagnosis were excluded from the study.

The institutional review board of the University of Gyeongsang National University Hospital approved this retrospective study and waived the requirement for informed consent because we used only deidentified data collected as part of clinical practice (IRB No. 2021-11-011-001).

### 2.2. DE-CTA

Angiography was conducted using a third-generation dual-source CT scanner (Somatom Force; Siemens Healthcare, Forchheim, Germany) with the following parameters applied: tube voltages, 150 and 100 kV; effective tube currents, 69 (150 kVp) and 124 mAs (100 kVp); gantry rotation time, 0.5 s; pitch factor, 0.45; automatic tube current modulation (CARE Dose 4D). Patients were in the supine position and were scanned feet first.

The same CA injection protocol was applied in all cases. The volume of CA was 1.5 mL/kg body weight (mean, 93 ± 18 mL). Typically, approximately 95 mL of iodinated CA (Pamiray, Dongkuk, Seoul Korea) was applied using a dual-syringe injector.

The flow rate was 5 mL/s for the first 30 mL of CA followed by a bolus of 90 mL CA and a saline flush of 40 mL at 3.5 mL/s. CT acquisition was initiated using the CARE-bolus technique when the average density within a region of interest was at the level of 150 HU during CA infusion of 80 mL with a delay of 5 s. The runoff scan covered the volume from the infrarenal aorta to the toes.

### 2.3. CT Image Post-Processing

CT images were reconstructed axially with a slice thickness of 1.5 mm and increments of 1 mm using filtered back-projection and a medium vessel kernel. Axial reconstructions were additionally post-processed through a DE application (Syngo via version VB 40B; Siemens Healthcare). The DE application eliminated bone and provided automated elimination of calcified plaques based on spectral differences from luminal iodine enhancement. Particular absorption properties at 100 and 150 kVp allowed differentiation of calcified structures and iodine. First, axial images were automatically combined to 100/150 kVp images with a weighting of 70%/30%. Axial MPRs with these settings were generated and used for evaluation. Second, three-dimensional MIPs after automatic bone and plaque removal were generated with a 180° circumference in 10° increments. These MIPs comprised a part of the image assessment. The 3D MIPs were sent to the picture archive and communication system (PACS) and were later evaluated by two CTA readers.

### 2.4. DSA

The mean time interval between the DSA acquisition and DE-CTA scan was 9.3 days (range: 0–30 days). Images were obtained using a Philips FD20 angiograph (Allura Xper FD20; Philips Corp., Eindhoven, the Netherlands). The DSA was performed by an interventional radiologist for planning of interventional or surgical therapy. The location of arterial access and extent of DSA imaging coverage were determined according to the clinical presentation and results of the previous CTA. In common cases with an infra-inguinal lesion limited to one lower extremity, an antegrade approach for the ipsilateral lower extremity runoff was chosen. In cases with iliac artery stenosis and if a crossover approach was judged appropriate, a retrograde femoral puncture was performed. Iodinated CA (iopamidol; Scanlux; Sanochemia, Vienna Austria) was manually injected, and posterior–anterior projection images were acquired using the stepping-table technique. Whenever necessary for correct stenosis quantification, added oblique projections were acquired. A variable frame rate with 1–2 images per second was used.

### 2.5. Image Analysis

The arterial tree was divided into 12 segments per patient (6 segments per lower extremity) for analysis. Paired lower-extremity segments were as follows: common iliac artery (CIA) to common femoral artery, superficial femoral artery (SFA), popliteal artery, anterior tibial artery (ATA), posterior tibial artery (PTA), and peroneal artery (PER). To ensure consistency between readers, the SFA, popliteal artery, ATA, PTA, and PER segments were separated using calipers by a non-reader vascular radiologist prior to image analysis. For the analysis of the DE-APR, including the automatically generated tool, only 3D MIPs were used for comparison with DSA and no axial images were used. DE-standard reconstruction (DE-SR), DE-APR MIP (APR), DE-SR with APR, and DSA images were independently reviewed by 2 consultant vascular/interventional radiologists on a PACS workstation (Impax v6.5.2; AGFA Healthcare, Morstel, Belgium) with windowing and multiplanar reconstruction capability. Readers were blinded to patient identity and presentation; the DE-SR, APR, DE-SR with APR, and DSA studies were isolated and randomized for review. Disagreements in reader DSA scores were resolved by consensus to provide a single reference standard.

For DE-SR, APR, and DE-SR with APR, the presence or absence of hemodynamically significant stenosis (≥50%) [18,22] was recorded for each arterial segment. Stenosis grading was usually done visually but the readers were free to use an electronic caliper in unclear cases. In cases with multiple arterial stenoses in a single segment, the highest grade of stenosis was evaluated. Each segment was also evaluated for the presence of calcification (0, no or minor calcifications that involved less than a third of the lumen circumference; 1, moderate or severe calcifications that involved more than a third of the lumen circumference). For DSA, segments were likewise evaluated for significant stenosis.

To evaluate the diagnostic accuracy of various imaging modalities, each dataset was evaluated by the two observers who had performed the individual image quality evaluation after a 6-week interval. Observers were asked to independently evaluate each segment for the presence of insignificant or significant stenosis using the previously mentioned approach. Discrepancies in the significance of a lesion was resolved during an agreement interpretation meeting 7 days later.

### 2.6. Statistical Analysis

Statistical analyses were performed using dedicated software (SPSS Statistics version 25; IBM Corp., Armonk, NY, USA; and R foundation for statistical computing, version 3.6.3, Vienna, Austria). Analysis was performed by region: AIS containing the CIA to EIA; FPS containing the SFA and popliteal artery; and BTKS containing the ATA, PTA, and PEA. There was considerable inter-reader agreement; therefore, the results were combined to determine the sensitivity, specificity, and accuracy of each image approach in the detection of significant stenosis (≥50%) using DSA as the reference standard. Accuracy was defined as the area under the receiver operating characteristic curve. Delong and bootstrap methods test were performed for differences in accuracy. Non-diagnostic DSA segments were excluded from the comparative analysis. For all pairwise comparisons, *p* < 0.05 was considered statistically significant.

## 3. Results

During the 24-month study period, 230 patients with PAD were referred for DE-CTA of the peripheral arteries. A total of 101 patients were excluded because there was a lack of endovascular treatment but surgical or conservative treatment was performed after DE-CTA. In the other 29 patients, endovascular treatment was not performed within 30 days after CTA. Hence, the final study included 100 patients (mean age: 70.4 years; range: 40–89 years; women: *n* = 17; men: *n* = 83). In four patients, DE-CTA and DSA scans were performed again during the study period due to repeated PAD symptoms. Figure 1 presents the flowchart of the patient-selection process. Patient demographics, including cardiovascular risk factors and PAD symptoms using the Fontaine classification, are presented in Table 1.

The DSA procedure was not specifically applied in this study. As a result, DSA did not represent the entire vasculature of each patient’s lower extremities but only the treatment-relevant areas, including inflow and outflow. As a result, interventions were reserved to one lower extremity in 55 patients. For the conventional bone-removal technique, two femoral segments and five popliteal segments could not be evaluated due to artifacts of a knee and femoral prosthesis.

### 3.1. Image Quality

A total of 530 CTA segments were reviewed. The image quality was considered “non-diagnostic” in 20 segments (3.8%), “fair” in 97 segments (18.3%), and “good” in 413 segments (77.9%) (Table 2). When comparing image quality between regions, a significant deterioration in image quality was observed between regions with larger arteries compared with those regions with progressively smaller caliber arteries. The reasons cited for non-diagnostic or fair image quality were venous contamination (30 segments), suboptimal contrast opacification (22 segments), and artifacts from metallic implants (7 segments).

### 3.2. Diagnostic Performance

The gold standard, DSA, revealed 258 segments without stenosis or mild stenosis and 272 segments with hemodynamically significant stenosis. Table 3 summarizes the diagnostic performance of various imaging approaches in the detection of significant stenosis and Table 4 lists the corresponding *p*-values for accuracy parallels. The sensitivity consistently exceeded 95% for all imaging approaches.

However, the specificity and accuracy varied depending on the use of APR. DE-SR with APR achieved specificity values >90%, whereas the other imaging approaches yielded specificity values less than 85%. The accuracy was also highest for DE-SR with APR (>95%), with significant differences for other imaging approaches.

Table 3 shows the diagnostic performance for vascular groups when the detection was considered positive for significant stenosis (≥50%) in at least one segment per group. Table 4 presents the corresponding *p*-values for the pairwise parallels. The sensitivity was generally >95% regardless of the vascular group. The specificity had a tendency to decrease from the AIS to BTKS according to vascular size when APR was used. DE-SR had the lowest specificity and accuracy value for the AIS group.

### 3.3. Impact of Vascular Wall Calcification

Of the 530 analyzed arterial segments, 211 (39.8%) segments showed no or mild vascular wall calcification and 319 (60.2%) segments had calcification of the lumen circumference. Table 5 presents the diagnostic performance for vascular wall calcification when the detection was considered positive for significant stenosis (≥50%) in at least one segment per group. Vascular wall calcification did not change the sensitivity for any DE-CTA images (DE-SR: 98.68% vs. 97.96%; APR: 98.68% vs. 97.45%; and DE-SR with APR: 98.68% vs. 97.96%). The specificity was markedly decreased in vascular calcification for all image approaches (DE-SR: 91.85% vs. 69.11%; APR: 87.41% vs. 78.86%; and DE-SR with APR: 94.81% vs. 78.43%). The accuracy was slightly lower in vascular calcification for all image approaches, especially DE-SR, which exhibited a considerable reduction in accuracy in calcified segments (DE-SR: 95.27% vs. 83.53%; APR: 93.05% vs. 88.16%; and DE-SR with APR: 96.75% vs. 93.70%). Figure 2 and Figure 3 present a comparison among DE-SR, APR, and DSA in two cases.

The degree of calcification did not significantly influence the accuracy in AIS and FPS when APR was used (*p* = 0.595 for AIS and *p* = 0.495 for FPS). However, the degree of calcification significantly influenced the accuracy in the BTKS (*p* = 0.002).

## 4. Discussion

DE-CTA has been reported to be useful in the diagnosis of lower-extremity arterial stenosis in patients with PAD. DE-CTA that employs the bone and plaque removal technique can improve the diagnostic accuracy [3,19,20,23]. Our findings revealed a clear improvement in the overall diagnostic performance with DE-SR with APR compared to DE-SR alone.

The reported sensitivity to detect hemodynamically significant stenosis in studies ranged from 84% [19] to 100% [24]; a contemporary systematic review on lower-extremity runoff DE-CTA reported a pooled sensitivity of 95.0% [21]. This was consistent with our results (97.79–98.16%). Vascular wall calcification and vascular lesion did not significantly affect the sensitivity in our study.

With regard to specificity, the results varied considerably among studies. A pooled specificity of 79.8% was reported [21], whereas other studies reported high specificity values of 94.1% [14] and 91.8% [24] with the bone and plaque removal technique; this was similar to our findings, which ranged from 81.01% (DE-SR) to 92.25% (DE-SR with APR).

To our knowledge, no published studies on DE-CTA using the bone and plaque removal technique for the lower extremities focused on the diagnostic performance solely based on the calcified lesion. We independently evaluated the diagnostic accuracy of the DE-APR technique for calcified lesions.

There was no significant difference in the diagnostic performance with the accuracy among DE-SR, APR, and DE-SR with APR in none-to-mild calcified stenotic lesions. After categorization into none-to-mild and moderate-to-severe calcification, the specificity of DE-SR was markedly reduced to <70% with a decreased accuracy, whereas the results of DE-SR with APR exhibited no significant reduction (94.81% to 89.43%) with a high accuracy (93.7%). Although calcification strongly affected the specificity of DE-CTA, APR improved the diagnostic performance by reducing the impact of calcification.

Regarding 3D MIP imaging, a few studies reported assessment of 3D reconstructed images that provided views similar to DSA images [19,20]. This could increase the acceptance of peripheral CTA by clinicians and provide greater convenience. Our results revealed a higher diagnostic performance with a greater than 90% accuracy in the AIS and FPS despite moderate to severe calcification grades. However, there was a relatively lower accuracy in the BTKS lesion (76%) compared with that of DE-SR (86.14%). Brockmann et al. reported a 90.9% accuracy in calf lesions [20]. However, the study population comprised 20 patients, which was a relatively low number to achieve statistically sufficient data; moreover, the data included non-calcification lesions. In our study, for the BTKS, overall accuracy increased to 87.32%; thus, there was selection bias if no or mild calcification lesions were involved.

Kau et al. compared 3D MIP with DSA in 58 patients [19] and reported overall sensitivity and specificity values of 84 and 67%, respectively, with a relatively lower accuracy for the BTKS. Our overall APR results, with a sensitivity of 97.79% and a specificity of 90.56%, exceeded the values from our previous study. There could be various reasons for these findings, but we considered the most important factor to be the automated bone and plaque removal. We evaluated the latest generation using the most recent version. Nevertheless, our results also showed the least specificity with the BTKS; however, our value increased to 75.59% for the BTKS compared with that of a previous study (51%). In this study, a good DE-CTA image-quality rate (57.9%) was lower than the image-quality rates of AIS and FPS (95.2% and 92.2%). We believe that image-quality differences such as venous contamination and suboptimal opacification impacted the 3D MIP image processing in addition to the fine vascular lumen in the BTKS.

This study had several limitations. First, only a single center was included and a relatively small number of patients were enrolled in this study. The reason for this was that the role of DE-CTA with APR in patients with PAD is still emerging, making it difficult to recruit more patients. This limitation could be overcome through more multicenter studies. Second, there was a selection bias due to the study’s retrospective nature and clinical routine setting, as only patients with DE-CTA findings suggestive of potentially treatment-requiring lesions were referred to undergo DSA, which might also limit the value of DSA as the gold standard. Third, in most cases, DSA was done unilaterally and only for a specific target area, which resulted in a lower number of matching abdominal aorta and lower-extremity arteries in DSA than those in DE-CTA. Fourth, as DE-CTA images were not optimized for some patients, there were images with a limited image quality, indicating vein contamination or suboptimal contrast opacification. This should be considered in-depth in a future study.

## 5. Conclusions

DE-SR with APR significantly increased the diagnostic accuracy for all lower-extremity segments to evaluate hemodynamically significant stenosis in patients with symptomatic PAD and transcended the effect of vascular wall calcification compared to DE-SR. Furthermore, APR exhibited favorable diagnostic performance in the aorto-iliac and femoro-popliteal segments with good agreement with DSA even in cases of moderate-to-severe vascular wall calcification.

## Figures and Tables

**Figure 1 medicina-58-01435-f001:**
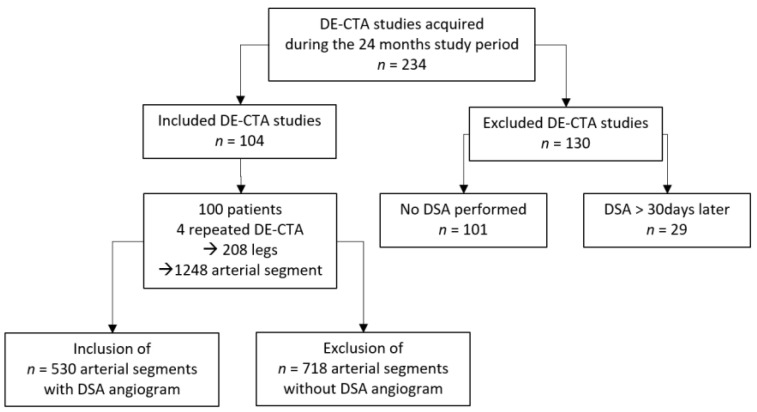
Patient selection flowchart.

**Figure 2 medicina-58-01435-f002:**
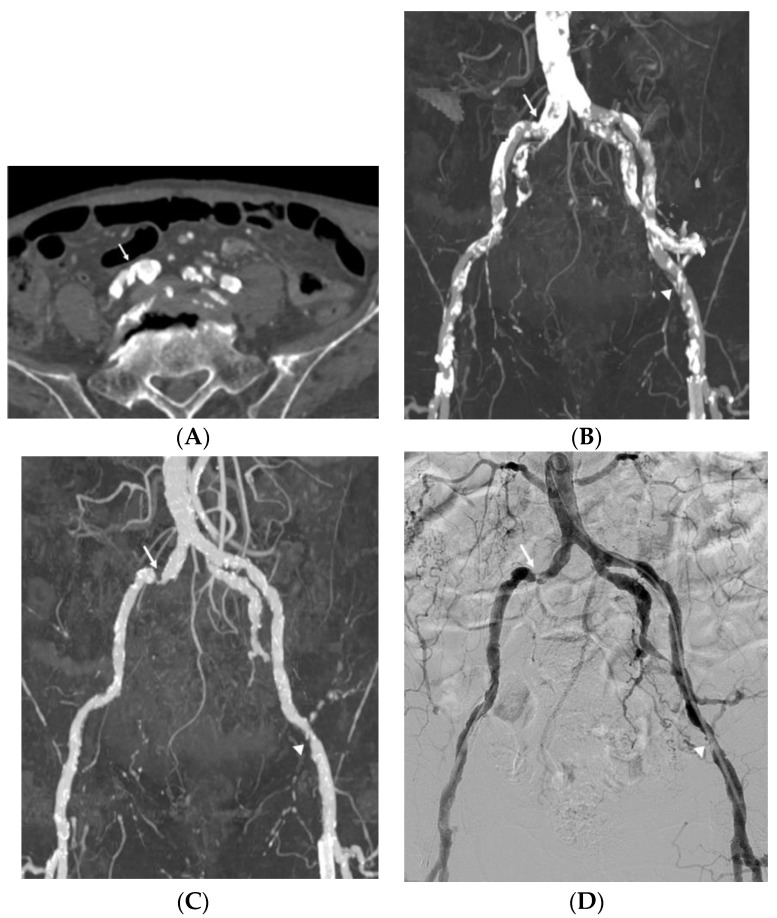
Imaging study of an 82-year-old man with claudication and calcified plaques in the right common femoral artery. Reliable stenosis evaluation was limited due to severe vascular wall calcification (arrow) in the standard reconstruction axial image (**A**) and maximum intensity projection (MIP) image (**B**). Dual-energy computed tomography (CT) automatic plaque removal MIP image (**C**) shows hemodynamically significant stenosis (arrow) confirmed by the digital subtraction angiography (DSA), which is the gold standard (**D**). Another hemodynamically significant stenosis (arrowhead) of the left external iliac artery on the automatic plaque removal MIP image also showed good correlation with DSA.

**Figure 3 medicina-58-01435-f003:**
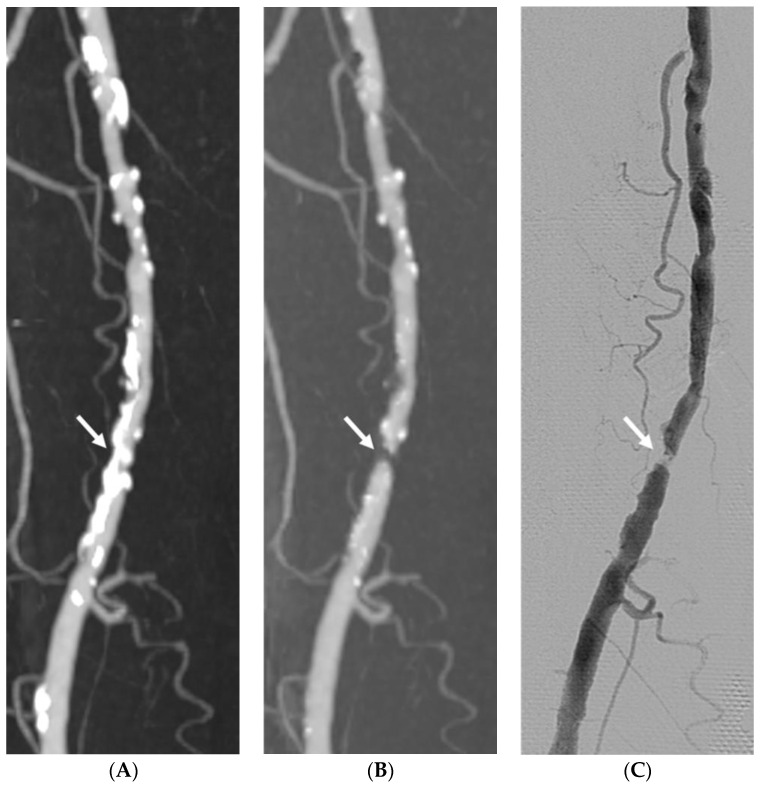
Imaging study of a 74-year-old man with claudication and severe vascular calcification in the right superficial femoral artery comparing dual-energy computed tomography (CT) standard reconstruction maximum intensity projection (MIP) image (**A**), automatic plaque removal MIP image (**B**), and the gold standard DSA (**C**). The automatic plaque removal maximum intensity projection (MIP) image reveals one hemodynamically significant stenosis (arrow) confirmed by DSA.

**Table 1 medicina-58-01435-t001:** Patient demographics (*n* = 100).

Sex	
Male	83 (83)
Female	17 (17)
Age (years)	70.4 [40, 89]
History	
DM	46 (46)
HTN	56 (56)
Hyperlipidemia	8 (8)
Smoking	67 (67)
CAD	18 (18)
CVD	11 (11)
CKD	12 (12)
Fontaine stage for chronic PAD	
I	14 (14)
IIA	13 (13)
IIB	10 (10)
III	48 (48)
IV	15 (15)
Previous PTA	18
Combined op	7
PTA time (days)	9.3 ± 13.06

Values are presented as mean (range), *n* (%), and mean ± standard deviation. DM, diabetes mellitus; HTN, hypertension; CAD, coronary artery disease; CVD, cardiovascular disease; CKD, chronic kidney disease; PAD, peripheral artery disease; PTA, percutaneous transluminal angioplasty.

**Table 2 medicina-58-01435-t002:** DE-CTA image quality.

Segment	Image Quality		
	0 (Non-Diagnostic)	1 (Fair)	2 (Good)
AIS	4 (3.1)	2 (1.6)	123 (95.3)
FPS	5 (3.0)	8 (4.8)	155 (92.2)
BTKS	11 (4.7)	87 (37.3)	135 (57.9)
Total	20 (3.8)	97 (18.3)	413 (77.9)

Values are presented as no. of segments (%). DE-CTA, dual-energy computed angiography; AIS, aorto-iliac segment; FPS, femoro-popliteal segment; BTKS, below-the-knee segment.

**Table 3 medicina-58-01435-t003:** Sensitivity, specificity, and accuracy according to arterial segment.

Segment	Sensitivity (%)	Specificity (%)	Accuracy (%) (95% CI)
AIS (*n* = 129)			
DE-SR	97.10	70.00	83.55 (77.37–89.73)
APR	95.65	93.33	94.49 (90.49–98.49)
DE-SR with APR	97.10	96.67	96.88 (93.85–99.92)
FPS (*n* = 168)			
DE-SR	98.97	80.28	89.63 (84.86–94.39)
APR	97.94	88.73	93.34 (89.37–97.30)
DE-SR with APR	99.97	91.55	95.26 (91.85–98.67)
BTKS (*n* = 233)			
DE-SR	98.11	86.61	92.36 (89.12–95.61)
APR	99.06	75.59	87.32 (83.46–91.19)
DE-SR with APR	98.11	90.55	94.33 (91.47–97.20)
Overall (*n* = 530)			
DE-SR	98.16	81.01	89.58 (87.06–92.11)
APR	97.79	83.33	90.56 (88.12–93.00)
DE-SR with APR	98.16	92.25	95.20 (93.39–97.02)

CI, confidence interval; AIS, aorto-iliac segment; DE-SR, dual-energy standard reconstruction image; ARP, dual-energy automatic plaque removal maximum intensity projection image; FPS, femoro-popliteal segment; BTKS, below-the-knee segment.

**Table 4 medicina-58-01435-t004:** *p*-Values for the pairwise comparison of accuracies in detection of significant stenosis (according to segment) presented in Table 3.

	DE-SR with APR				APR			
	AIS	FPS	BTKS	Overall	AIS	FPS	BTKS	Overall
DE-SR	**<0.001**	**0.003**	**0.023**	**<0.001**	**0.001**	0.065	**0.012**	0.495
APR	0.082	0.084	**<0.001**	**<0.001**				

Significant values are in boldface. DE-SR, dual-energy standard reconstruction image; ARP, dual-energy automatic plaque removal maximum intensity projection image; AIS, aorto-iliac segment; FPS, femoro-popliteal segment; BTKS, below-the-knee segment.

**Table 5 medicina-58-01435-t005:** Sensitivity, specificity, and accuracy according to vascular wall calcification and arterial segment.

Segment	Sensitivity (%)	Specificity (%)	Accuracy (%)
No or mild calcification			
AIS (*n* = 15)			
DE-SR	100.00	90.00	95.00
APR	100.00	80.00	90.00
DE-SR with APR	100.00	100.00	100.00
FPS (*n* = 64)			
DE-SR	96.00	92.31	94.15
APR	96.00	92.31	94.15
DE-SR with APR	96.00	94.87	95.44
BTKS (*n* = 132)			
DE-SR	100.00	91.86	95.93
APR	100.00	86.05	93.02
DE-SR with APR	100.00	94.19	97.09
Overall (*n* = 211)			
DE-SR	98.68	91.85	95.27
APR	98.68	87.41	93.05
DE-SR with APR	98.68	94.81	96.75
Moderate or severe calcification			
AIS (*n* = 114)			
DE-SR	96.88	66.00	81.44
APR	95.31	96.00	95.66
DE-SR with APR	95.31	96.00	96.44
FPS (*n* = 104)			
DE-SR	100.00	65.63	82.81
APR	98.61	94.38	91.49
DE-SR with APR	100.00	87.50	93.75
BTKS (*n* = 101)			
DE-SR	96.67	75.61	86.14
APR	98.33	53.66	76.00
DE-SR with APR	96.67	82.93	89.80
Overall (*n* = 319)			
DE-SR	97.96	69.11	83.53
APR	97.45	78.86	88.16
DE-SR with APR	97.96	89.43	93.70

DE-SR, dual-energy standard reconstruction image; ARP, dual-energy automatic plaque removal maximum intensity projection image; AIS, aorto-iliac segment; FPS, femoro-popliteal segment; BTKS, below-the-knee segment.

## Data Availability

All relevant data are contained within the paper.

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
