# Peer review of "Automatic Plaque Removal Using Dual-Energy Computed Tomography Angiography: Diagnostic Accuracy and Utility in Patients with Peripheral Artery Disease"

_medicina, 2022, doi:10.3390/medicina58101435_

Round 1

Reviewer 1 Report

I have received the manuscript entitled: “Automatic plaque removal using dual-energy computed tomography angiography: diagnostic accuracy and utility in patients with peripheral artery disease” prepared by Koo et al. and submitted to the Medicina (IF-2,948) to prepare the review comments and opinion. Firstly, it should be emphasized that the subject studied in this publication, such as diagnostic tools used in cardiovascular diseases, including peripheral arterial disease, is extremely important because cardiovascular diseases are one of the most important causes of morbidity and mortality worldwide, so improvement in diagnosis may lead to a better quality of medical care over patients with cardiovascular diseases. Secondly, the paper is generally well prepared and it presents quite high scientific value, and it should be considered for publication in the Medicina in the future, but in my opinion, some significant improvements and small corrections are necessary to improve the value of this paper further, which I describe below.

1)     In the abstract, there is the following sentence “In the three main arterial segments (…)”. Although it is obvious and is described in the main text, it should be explained also in the abstract what three segments do the Authors think about.

2)     The introduction is too laconic, in my opinion. It is worth writing some words about the peripheral arterial disease (PAD) in general. PAD is one of the clinical manifestations of atherosclerotic cardiovascular disease (ACVD). ACVD is one of the most importancausesse of mortality and morbidity worldwide. Endothelial dysfunction is one of the most important processes in the pathogenesis of ACVD. It should be highlighted that atherosclerosis is the most common cause of PAD. It should be mentioned that diabetes is a significant risk factor for the development of PAD and on the other hand, diabetes affects the course of PAD: in patients with PAD and coexisting diabetes, the lesions are more often multilevel as well as arteries below the knee are more often affected, and arteries’ walls are usually more calcified. PAD in patients with diabetes is associated with an increased risk of frailty syndrome development. Endovascular treatment plays a significant role in the treatment of patients with severe PAD, but restenosis is associated with the risk of diminished effectiveness and the necessity of reintervention. It is also worth noting that assessment of arterial stiffness and endothelial dysfunction are concerning the increasing interest in research as well as in clinical practice. (See for example: 10.3390/ijerph17249339; 10.3390/ijerph191811242).

3)     It should be not “lower knee segment” abbreviated as “LKS” but “below-the-knee” abbreviated as “BTK” or eventually “below-the-knee segment” abbreviated as “BTKS” to maintain the same convention as in other segments. (lines 51-52 and further parts of the text).

4)     Was this study a retrospective analysis of data from medical documentation or a prospective study? If prospective, was the research project assessed by Ethical Committee? The number of the document with the potential permission of the Ethical Committee should be written in the text. If retrospective, it should be emphasized. I see that in “limitations of the study” the Authors mentioned the retrospective character of the study, but it should be elucidated in material and methods.

5)     In my opinion, the Authors should avoid using the word “leg”. “Extremity” or “lower extremity” should be used.

6)     In the description of statistical methods, the Authors indicated p < 0.05 to be significantly important, but there is no information about statistical tests used to test statistical hypotheses. Please, add some information about it. Have the data presented as means and standard deviations been assessed to be consistent with the normal distribution? If so, in what way? If not, why was it decided to choose such measures and not to choose the median and standard deviations?

7)     I suggest not to use abbreviations in table 1 apart from PAD and PTA. Other abbreviations are unnecessary. In table 1, there is “previous PTA”, so I understood it as “previous percutaneous transluminal angioplasty” whereas under the table, the abbreviations PTA has been explained as “posterior tibial artery”. Please, check it.

8)     The list of references should be prepared in accordance with the MDPI rules.

Author Response

Thank you for your review regarding our manuscript medicina-1922764, entitled “Automatic plaque removal using dual-energy computed tomography angiography: diagnostic accuracy and utility in patients with peripheral artery disease”

   We wish to thank you for your valuable comments and helpful suggestions, which contributed significantly to the improvement of our manuscript. We have responded to the comments of the reviewers in a point-by-point fashion. 

Reviewer 2 Report

This article evaluates the accuracy of dual-energy automatic plaque removal in patients with symptomatic peripheral arterial disease. 

The study is well conducted. Methods and results are fairly stated. However, I have one minor suggestion. 

I would improve the introduction. More in-depth information on the different evaluation methods for PAD screening (i.e a table of comparisons between the different methods). How is MIP generation performed, is not clear to me? I am not an expert in the field of CTA. More information is needed.  Also I would include which patients are prone to suffer from PAD as well as what is their clinical outcome. To further understate the need for a good screening method.

Author Response

Thank you for your review regarding our manuscript medicina-1922764, entitled “Automatic plaque removal using dual-energy computed tomography angiography: diagnostic accuracy and utility in patients with peripheral artery disease”

    We have changed the manuscript as much as possible according to the reviewer’s comments. We wish to thank you for your valuable comments and helpful suggestions, which contributed significantly to the improvement of our manuscript. We have responded to the comments of the reviewers in a point-by-point fashion. 

Round 2

Reviewer 1 Report

The work has some cognitive value and meets sufficient requirements in terms of its structure and preparation of the manuscript, but I believe that the manuscript would be more attractive if the authors followed the suggestions in my first review to a greater extent, especially in terms of extending the information contained in the introduction and increase in the number of references (22 items is very little).

Author Response

On behalf of the authors, I would like to thank you for the valuable and helpful comments on our submitted manuscript, medicina-1922764, entitled “Automatic plaque removal using dual-energy computed tomography angiography: diagnostic accuracy and utility in patients with peripheral artery disease”

We have carefully reviewed the comments and have revised the manuscript accordingly.

We would be happy to make further corrections if necessary, and look forward to hearing from you soon.

" Please see the attachment."
